# Huisgen [3 + 2] Dipolar Cycloadditions of Phthalazinium Ylides to Activated Symmetric and Non-Symmetric Alkynes

**DOI:** 10.3390/molecules25194416

**Published:** 2020-09-25

**Authors:** Vasilichia Antoci, Costel Moldoveanu, Ramona Danac, Violeta Mangalagiu, Gheorghita Zbancioc

**Affiliations:** 1Faculty of Chemistry, Alexandru Ioan Cuza University of Iasi, 11 Carol 1st Bvd, 700506 Iasi, Romania; vasilichia.antoci@uaic.ro (V.A.); costel.moldoveanu@uaic.ro (C.M.); rdanac@uaic.ro (R.D.); 2Institute of Interdisciplinary Research-CERNESIM Centre, Alexandru Ioan Cuza University of Iasi, 11 Carol I, 700506 Iasi, Romania

**Keywords:** Huisgen [3 + 2] dipolar cycloaddition, ylides, regiochemistry, selectivity, microwave, ultrasound, environmentally friendly, pyrrolophthalazine

## Abstract

We present herein a straightforward and efficient pathway for the synthesis of pyrrolophthalazine cycloadducts via Huisgen [3 + 2] dipolar cycloaddition reactions of phthalazinium ylides to methyl propiolate or dimethyl acetylenedicarboxylate (DMAD). A thoroughly comparative study concerning the efficiency of synthesis, conventional thermal heating (TH) versus microwave (MW) and ultrasound (US) irradiation, has been performed. The cycloaddition reactions of phthalazinium ylides to methyl propiolate occur regiospecific, with a single regioisomer being obtained. Under conventional TH, the cycloaddition reaction of phthalazinium ylides with DMAD occurs to a mixture of inseparable partial and fully aromatized pyrrolophthalazine cycloadducts, while MW or US irradiation are leading only to fully aromatized compounds, with the reactions becoming selective. A feasible mechanism for formation of fully aromatized compounds is presented. Besides selectivity, it has to be noticed that the reaction setup under MW or US irradiation offer a number of other certain advantages: higher yields, decreasing of the amount of used solvent comparative with TH, decreasing of the reaction time from hours to minutes and decreasing of the consumed energy; consequently, these reactions could be considered environmentally friendly.

Academic Editor: Gianfranco Favi

## 1. Introduction

Azaheterocyclic compounds, especially 1,2-diazine derivatives, constantly raise interest in the scientific community because they have shown potential utilizations for optoelectronics (fluorescent materials, sensors and biosensors, etc.) [1,2,3], medicinal chemistry (including anticancer [4,5,6,7,8], antituberculosis [9,10], anti-Human Immunodeficiency Virus (HIV) [11], antioxidant [12] and antimicrobial activity [8]), etc.

The Huisgen [3 + 2] dipolar cycloaddition reactions of cycloimmonium ylides to variously symmetric and non-symmetric substituted dipolarophiles, represent a facile and convenient way to obtain fused azaheterocycles compounds, which are otherwise very difficult or even impossible to obtain [13,14]. These reactions are still widely discussed because of their theoretical and practical interest. Thus, the cycloaddition reactions of pyridazinium [15,16,17], pyrimidinium [18] and phthalazinium ylides [19,20,21] with symmetrical substituted dipolarophiles raise interesting problems of stereochemistry, while the reaction of ylides with non-symmetrical dipolarophiles raises supplementary issues regarding regiochemistry and chorochemistry, problems which are widely discussed and far away from being solved.

In recent years, microwave- and ultrasound-assisted reactions have proved to be a new trend in synthetic organic chemistry [22,23,24,25,26], offering a facile and useful alternative in a large variety of syntheses [2,22,27,28,29,30,31,32,33,34,35]. In comparison with conventional thermal heating (TH), both microwave and ultrasound irradiation have several important advantages: improved yields, high purity of the compounds, increased selectivity, decreases of reaction time, lower costs and simplicity in handling and processing. Considering all these advantages, reactions under MW and US irradiation could be considered environmentally friendly [36,37].

In continuation of our efforts for synthesis of new azaheterocycles via Huisgen [3 + 2] cycloadditions reactions [1,2,12,20,22,27,32], we decided to study the synthesis of fused pyrrolophthalazine derivatives under conventional TH, MW and US irradiation. In equal measure, we were interested in developing new environmentally friendly methods for preparation of these derivatives using MW and US technologies.

## 2. Results and Discussion

In previous research work, we studied the synthesis under conventional thermal heating and under US or MW irradiation of a series of new pyridazinium/phthalazinium derivatives with a dihydroxyacetophenone skeleton as well as their antimicrobial and/or anticancer activity [32,34,38]. Considering the good results obtained, we decided to broaden these studies to the pyrrolophthalazine derivatives with a dihydroxyacetophenone skeleton.

The general approach used for the synthesis of pyrrolophthalazine derivatives, **5** and **6**, is straightforward and efficient and involves two steps: the in situ generation of the phthalazinium ylides with dihydroxyacetophenone skeleton **2a**–**e** (from the corresponding phthalazinium salts **1a**–**e**, in the presence of triehtylamine (TEA) as base), followed by a Huisgen [3 + 2] dipolar cycloaddition of phthalazinium ylides **2a**–**e** to the corresponding dipolarophiles (methyl propiolate or DMAD), Scheme 1.

Initially, we performed these reactions under conventional TH, in refluxing chloroform, with TEA (diluted in chloroform) being added dropwise at the beginning of the reaction. After 6 h of reflux, the reaction was stopped, and the products were isolated from the reaction mixture. The reaction of phthalazinium ylides (**2a**–**e**) with methyl propiolate leads to the fully aromatized cycloadducts **5a**–**e**, in moderate to good yields (48% to 76%, see Table 1).

Theoretically, the Huisgen [3 + 2] dipolar cycloadditions reactions of cycloimmonium ylides (dipole) to methyl propiolate (non-symmetrical substituted dipolarophile) raises problems of regiochemistry, because the double sense of addition of ylide to dipolarophile has often been found, with the formation of two regioisomers, **A** and **B** (Scheme 2), according to the orbital, steric and electronic factors [13,14,15,16,17,18,19,20].

As can be seen in Scheme 1, we find that in the case of the reaction of phthalazinium ylides **2a**–**e**, the bond is formed between the ylide carbon and the non-substituted carbon atom of the methyl propiolate regioisomer **A** (Scheme 2). The assignment of the structure of the compounds **5a**–**e** was made based on the spectral analysis (NMR-see Appendix A). In the HMBC long-range correlation spectra of the compound **5d** (see Appendix A), used as example, the signal of the proton from the second position gives correlation with the signal of the keto-carbonil from the eleventh position and with the quaternary carbon atom from the 10b position. Thus, the only possible structure of the compound **5d** is the **A** regioisomer. With the structure being in accordance with the electronic effects exerted in the methyl propiolate molecule, a single regioisomer is obtained, which means that the reaction is regiospecific under a charge control.

The reaction of phthalazinium ylides (**2a**–**e**) with DMAD gives a mixture of inseparable partial and fully aromatized compounds (**6a**–**e** and **7a**–**e**, respectively). Regardless of the separation methods we employed, we isolated in each case a mixture of compounds **6a**–**e** and **7a**–**e**. As to the mechanism, the formation of compounds **7a**–**e** indicate that the reaction occurs as a normal Huisgen [3 + 2] dipolar cycloaddition, leading in a first step to the unsaturated dihydropyrrolophthalazine cycloadducts **7′a**–**e**. In these cycloadducts, the hydrogen atom from the third position has an increased acidity due to the strong withdrawing effect of both the keto-carbonyl group from the eleventh position and of the highly electronegative nitrogen atom from the fourth position. Thus, under the influence of the base (triethylamine) which abstracts the acidic proton, the generated carbanion isomerizes, then the conjugated acid of the triethylamine transfers back the proton in the first position, leading to the dihydropyrrolophthalazine cycloadducts **7a**–**e**. Finally, via an oxidative dehydrogenation, the presence of a hydrogen acceptor which is most often oxygen from the air [18,39] leads to the thermodynamically more stable aromatized pyrrolophthalazine derivatives **6a**–**e** (Scheme 3). There are situations in which the aromatization can be accelerated by the use of oxidizing agents [40,41]. These experiments really showed that aromatization is delayed or even stopped in the nitrogen atmosphere and accelerated in the oxygen atmosphere. The oxidative dehydrogenation of the **7a**–**e** to **6a**–**e** cycloadducts occurs in the reaction mixture both before and after the workup of the reaction mixture during separation. 

The major disadvantages of these reactions under conventional TH are the lack of selectivity for the DMAD cycloaddition, long reaction time (360 min in case of reaction with methyl propiolate and one week in case of reaction with DMAD) and the high energy consumption of the synthesis carried out under conventional conditions.

As an alternative route, we have performed the synthesis of the pyrrolophthalazine derivatives with dihydroxyacetophenone skeleton under MW and US irradiation. For MW irradiation reactions, we used a monomode reactor (Monowave 300; Anton Paar, Graz, Austria). This reactor allows stirring of the reaction mixture (0 to 1200 rpm) and can reach up to 300 °C with temperature control. The reactions take place in a closed vessel at 30 bars of maximum pressure. In our case, the optimal reaction conditions were found to be 130 °C and 5.5–6 bars, with the reactions being completed after 10 to 40 min of irradiation. For US irradiation reactions, we used a reactor Bandelin (Sonopuls GM 3200, Berlin, Germany), with a nominal power of 200 W. The instrumentation used allowed us to control the pulse sequence, as well as the amplitude (mean percent of the nominal power) and the irradiation time. All these parameters are expected to influence the reaction. The Huisgen [3 + 2] dipolar cycloaddition was performed using 80% of the instrument nominal power and was completed after 20 to 60 min of irradiation.

The best results obtained with optimized reaction conditions under MW and US irradiation are summarized in Table 1, compared to the optimized reaction conditions used in thermal heating (TH) conditions.

As indicated in Table 1, under MW and US irradiation, the cycloaddition reactions of phthalazinium ylides with DMAD became selective, with only fully aromatized compounds (**6a**–**e**) being obtained. We may also notice that the use of MW and US irradiation induced a remarkable acceleration for the Huisegen [3 + 2] dipolar cycloaddition reaction, with the reaction times decreasing from 6 h to 10 min (under MW irradiation) or 20 min (under US irradiation) in the case of the reaction with methyl propiolate, while in the case of the reaction with DMAD, the decreasing is even more accentuated from one week to 40 min (under MW irradiation) or 60 min (under US irradiation). This acceleration of the cycloaddition reactions under unconventional heating may be attributed to the fact that both MW and US irradiation make it possible to reach much higher temperatures (higher than the boiling point of the solvent) than in the case of the thermal heating. We also wish to point out that under MW and US irradiation, the yields were higher (by 30%) and the solvent amounts used in the former were at least three times lower than the corresponding quantities used under conventional conditions (see the Materials and Methods). Consequently, the cycloaddition reaction under MW and US irradiation could be considered environmentally friendly.

Based on our previous results [30], we presume that the MW heating approach is more effective in [3 + 2] dipolar cycloaddition reactions due to two factors: the mode of action under MW irradiation and the structure of the ylide intermediate. The MW theory [24,30,31] states that the dielectric heating effect of MW depends essentially on the dipole moment of the molecules: the greater the dipole moment is, the larger the effect of the MW energy will be. The ylides are excellent dipoles (they have a 1,2-dipolar structure) and, therefore, the efficiency of MW heating increases considerably when compared with TH.

Concerning the US irradiation efficiency in the Huisgen [3 + 2] dipolar cycloaddition reaction, we presume that this efficiency was due to the cavitations phenomena, with the energy being more efficiently transmitted to the substrates compared to the reactions performed under conventional conditions. Also, the collapse of bubbles induced mechanical stress that could be transmitted to a target single bond, with this phenomenon being specific to ultrasound action [25].

## 3. Materials and Methods

### General Procedure

All the reagents and solvents were purchased from commercial sources and used without further purification. Melting points were recorded on an Electrothermal MEL-TEMP apparatus (Barnstead International, Dubuque, Iowa, USA) in open capillary tubes and are uncorrected. Analytical thin-layer chromatography was performed with commercial silica gel plates 60 F254 (Merck, Merck Darmstadt, Germany) and visualized with UV light. The NMR spectra were recorded on a Bruker Avance III 500 MHz spectrometer (Bruker, Vienna, Austria) operating at 500 MHz for ^1^H and 125 MHz for ^13^C. The following abbreviations were used to designate chemical shift multiplicities: s = singlet, d = doublet, t = triplet, m = multiplet. Chemical shifts were reported in delta (δ) units, part per million (ppm) and coupling constants (*J*) in Hz. Infrared (IR) data were recorded as films on potassium bromide (KBr) pellets on a Fourier-transform infrared spectroscopy (FT-IR) Shimadzu Prestige 8400 s spectrophotometer (Shimadzu, Kyoto, Japan). Ultrasound-assisted reactions were carried out using a Bandelin Ultrasound reactor (Sonopuls GM 3200, Bandelin electronic GmbH & Co. KG, Berlin, Germany), with a nominal power of 200 W and a frequency of 20 kHz. The booster horn SH 213 G (Bandelin electronic GmbH & Co. KG, Berlin, Germany) was fixed tightly to the ultrasonic converter. The titanium flat probe tip TT13 (diameter: 12.7 mm, length: 7 mm, Bandelin electronic GmbH & Co. KG, Berlin, Germany) was fixed tightly to the booster horn. The titanium probe tip was immersed in the used solvent. Microwave-assisted reactions were carried out using a monomode reactor Monowave 300 (Anton Paar, Graz, Austria). Some additional specifications of Monowave 300 (Graz, Austria) are as follows: microwave power 850 W, operation limits at 300 °C and 30 bars, reaction vial Borosilicat (Graz, Austria), operation volume between 0.5 and 20 mL, pressure control by hydraulic system, agitation with an integrated magnetic stirrer (0 to 1200 rpm) and cooling with compressed air.

#### General Procedure for Synthesis of Pyrrolophthalazine Derivatives, **5a**–**e** and **6a**–**e** under Conventional TH Conditions, MW and US Irradiation

A mixture of phthalazinium salts **1a**–**e** (1 mmol, 0.505 g) and DMAD (0.17 mL, 1.4 mmol) or methyl propiolate (0.12 mL, 1.4 mmol) was suspended in 20 mL chloroform. Then, triethylamine (0.19 mL, 3.5 mmol) dissolved in 10 mL chloroform was added dropwise under stirring for one hour. The stirring and refluxing were continued for 6 h to 1 week. After the reaction was finished Thin-layer chromatography (TLC), the obtained solution was cooled down at room temperature and then the reaction mixtures were washed with water (3 × 30 mL), dried over magnesium sulfate and evaporated under reduced pressure to give the crude product. The purification of the crude product was done by column chromatography on silica gel (eluted with CH_2_Cl_2_ to 98/2 CH_2_Cl_2_/CH_3_OH).

Under MW irradiation, the mixture of reagents was dissolved in 10 mL chloroform, placed in the reaction vessel (borosilicate) and exposed to microwave irradiation (from 10 min to 40 min; see Table 1). Using MW irradiation, the best results were obtained using a “temperature control” method. The “temperature control” method ensures a constant temperature (in this case 130 °C) and varying the power and pressure. This method takes place in three stages. In the first step, the temperature is raised as quickly as possible (within less than 1 min) by applying the maximum power. In the second step, the reaction mixture is kept at a constant temperature with the control of the magnetron power. In the last step, the reaction tube is cooled to 55 °C by stopping the irradiation and blowing the reaction vial with compressed air. Once the heating cycle is completed, the reaction vial is removed from the reactor, and processed as indicated for TH.

Under US irradiation, the mixture of reagents (in 10 mL chloroform) was placed into the reaction vessel and exposed to irradiation (from 20 min to 60 min; see Table 1). Once the irradiation cycle was completed, the reaction tube was removed from the reactor, and processed as indicated above for TH.

Dimethyl 2,2′-((4-(1-(methoxycarbonyl)pyrrolo[2,1-a]phthalazine-3-carbonyl)-1,3-phenylene)bis(oxy)) diacetate (**5a**). 0.375 g, 74% (under classical heating), 0.471 g, 93% (under microwave) and 0.456 g, 90% (under ultrasounds) as white crystals, m.p. 135–136 °C; R_f_ (98/2 CH_2_Cl_2_/CH_3_OH) 0.36; IR (cm^−1^): 3111, 3057, 3026 (C-H arom.), 2951 (C-H aliph.), 1751, 1707 (C=O, ester), 1641 (C=O, keto), 1566, 1531, 1479, 1371 (aromatic and heteroaromatic ring), 1264, 1233, 1199, 1108 (C-O-C, ester); ^1^H-NMR (500 MHz, CDCl_3_): δ 9.82 (1H, d, *J =* 8.5 Hz, H-10); 8.67 (1H, s, H-6); 7.87 (2H, m, overlapped peaks, H-7, H-9), 7.77 (1H, s, H-2), 7.72 (1H, t, *J =* 7.5 Hz, H-8), 7.58 (1H, d, *J =* 8.5 Hz, H-17), 6.58 (1H, dd, *J =* 2.0, 8.5 Hz, H-16), 6.48 (1H, d, *J =* 2.0 Hz, H-14), 4.67 (2H: CH_2_ of methyl acetate group from 15 position, s), 4.44 (2H: CH_2_ of methyl acetate group from 13 position, s), 3.92 (3H, s, CH_3_ of methoxycarbonyl group from 1 position), 3.77 (3H: CH_3_ of methyl acetate group from 15 position, s), 3.70 (3H: CH_3_ of methyl acetate group from 13 position, s), ^13^C-NMR (125 MHz, CDCl_3_): δ 182.8 (CO keto), 168.3 (CO keto ester from 15 position), 168.0 (CO keto ester from 13 position), 164.9 (CO keto ester from 1 position), 161.2 (C-15), 157.8 (C-13), 146.0 (C-6), 132.9 (C-9), 132.1 (C-17), 129.7 (C-10b), 129.5 (C-8), 129.2 (C-3), 127.5 (C-7), 127.4 (C-10), 126.9 (C-12), 124.8 (C-10a), 124.7 (C-2); 122.0 (C-6a), 108.1 (C-1), 106.2 (C-16), 101.4 (C-14), 66.4 (CH_2_ of methyl acetate group from 15 position, -CH_2_-COOMe), 65.5 (CH_2_ of methyl acetate group from 13 position, -CH_2_-COOMe), 53.8 (CH_3_ of methyl acetate group from 15 position, -CH_2_-COOMe), 52.3 (CH_3_ of methyl acetate group from 13 position, -CH_2_-COOMe), 51.8 (CH_3_ of methoxycarbonyl group from 1 position). Anal. calc. for C_26_H_22_N_2_O_9_ (506.47): C 61.66, H 4.38, N 5.53; found: C 61.62, H 4.33, N 5.46.

Dimethyl 2,2′-((2-(1-(methoxycarbonyl)pyrrolo[2,1-a]phthalazine-3-carbonyl)-1,4-phenylene)bis(oxy)) diacetate (**5b**) 0.349 g, 69% (under classical heating), 0.451 g, 89% (under microwave) and 0.430 g, 85% (under ultrasounds) as yellowish crystals, m.p. 1′′24–125 °C; R_f_ (98/2 CH_2_Cl_2_/CH_3_OH) 0.37; IR (cm^−1^): 3106, 3066, 3001 (C-H arom.), 2955 (C-H aliph.), 1748, 1703 (C=O, ester), 1633 (C=O, keto), 1549, 1519, 1497, 1442, 1397 (aromatic and heteroaromatic ring), 1289, 1250, 1203, 1184, 1131 (C-O-C, ester); ^1^H-NMR (500 MHz, CDCl_3_): δ 9.83 (1H, d, *J =* 8 Hz, H-10); 8.75 (1H, s, H-6); 7.90 (2H, t, *J =* 8, 7.5 Hz, H-7, H-9), 7.76 (1H, t, *J =* 7.5 Hz, H-8), 7.74 (1H, s, H-2), 7.09 (1H, d, *J =* 2.5 Hz, H-17), 7.04 (1H, dd, *J =* 2.5, 9.0 Hz, H-15),6.88 (1H, d, *J =* 9.0 Hz, H-14), 4.64 (2H: CH_2_ of methyl acetate group from 16 position, s), 4.50 (2H: CH_2_ of methyl acetate group from 13 position, s), 3.92 (3H, s, CH_3_ of methoxycarbonyl group from 1 position), 3.80 (3H: CH_3_ of methyl acetate group from 16 position, s), 3.61 (3H: CH_3_ of methyl acetate group from 13 position, s), ^13^C-NMR (125 MHz, CDCl_3_): δ 183.0 (CO keto), 169.3 (CO keto ester from 16 position), 169.2 (CO keto ester from 13 position), 166.6 (CO keto ester from 1 position), 152.8 (C-16), 150.9 (C-13), 146.5 (C-6), 133.1 (C-9), 132.2 (C-10b), 130.5 (C-3), 130.0 (C-8), 128.2 (C-12), 127.7 (C-7), 127.6 (C-10), 126.9 (C-10a), 126.2 (C-2); 122.3 (C-6a), 118.6 (C-15), 116.0 (C-17), 115.8 (C-14), 108.6 (C-1), 67.7 (CH_2_ of methyl acetate group from 16 position, -CH_2_-COOMe), 66.2 (CH_2_ of methyl acetate group from 13 position, -CH_2_-COOMe), 52.4 (CH_3_ of methyl acetate group from 16 position, -CH_2_-COOMe), 52.2 (CH_3_ of methyl acetate group from 13 position, -CH_2_-COOMe), 52.0 (CH_3_ of methoxycarbonyl group from 1 position). Anal. calc. for C_26_H_22_N_2_O_9_ (506.47): C 61.66, H 4.38, N 5.53; found: C 61.61, H 4.34, N 5.47.

Dimethyl 2,2′-((2-(1-(methoxycarbonyl)pyrrolo[2,1-a]phthalazine-3-carbonyl)-1,3-phenylene)bis(oxy)) diacetate (**5c**). 0.334 g, 66% (under classical heating), 0.436 g, 86% (under microwave) and 0.430 g, 85% (under ultrasounds) as yellowish crystals, m.p. 145–146 °C; R_f_ (98/2 CH_2_Cl_2_/CH_3_OH) 0.36; IR (cm^−1^): 3100, 3064, 3001 (C-H arom.), 2957 (C-H aliph.), 1748, 1703 (C=O, ester), 1649 (C=O, keto), 1553, 1503, 1488, 1435, 1371 (aromatic and heteroaromatic ring), 1272, 1250, 1213, 1115 (C-O-C, ester); ^1^H-NMR (500 MHz, CDCl_3_): δ 9.73 (1H, d, *J =* 8.5 Hz, H-10); 8.79 (1H, s, H-6); 7.83 (2H, q, *J =* 7.5, 8.5, Hz, H-7, H-9), 7.74 (1H, s, H-2), 7.69 (1H, t, *J =* 7.5 Hz, H-8), 7.28 (1H, t, *J =* 8.5 Hz, H-15), 6.53 (2H, d, *J =* 8.5 Hz, H-14, H-16), 4.58 (4H: CH_2_ of methyl acetate groups from 13 and 17 positions, s), 3.87 (3H, s, CH_3_ of methoxycarbonyl group from 1 position), 3.64 (6H: CH_3_ of methyl acetate groups from 13 and 17 positions, s), ^13^C-NMR (125 MHz, CDCl_3_): δ 181.5 (CO keto), 169.1 (CO keto ester from 13 and 17 positions), 164.7 (CO keto ester from 1 position), 156.2 (C-13, C-17), 146.6 (C-6), 132.8 (C-9), 130.7 (C-15), 130.6 (C-10b), 129.8 (C-8), 128.1 (C-3), 127.5 (C-7), 127.4 (C-10), 126.8 (C-2), 126.6 (C-10a), 122.2 (C-6a), 120.8 (C-12), 108.4 (C-1), 106.1 (C-14, C-16), 66.0 (CH_2_ of methyl acetate groups from 13 and 17 positions, -CH_2_-COOMe), 52.1 (CH_3_ of methyl acetate groups from 13 and 17 positions, -CH_2_-COOMe), 51.8 (CH_3_ of methoxycarbonyl group from 1 position). Anal. calc. for C_26_H_22_N_2_O_9_ (506.47): C 61.66, H 4.38, N 5.53; found: C 61.59, H 4.35, N 5.45.

Dimethyl 2,2′-((4-(1-(methoxycarbonyl)pyrrolo[2,1-a]phthalazine-3-carbonyl)-1,2-phenylene)bis(oxy)) diacetate (**5d**). 0.355 g, 70% (under classical heating), 0.456 g, 90% (under microwave) and 0.456 g, 90% (under ultrasounds) as white crystals, m.p. 132–133 °C; R_f_ (98/2 CH_2_Cl_2_/CH_3_OH) 0.40; IR (cm^−1^): 3103, 3044, 3004 (C-H arom.), 2952 (C-H aliph.), 1751, 1705 (C=O, ester), 1638 (C=O, keto), 1554, 1535, 1498, 1453, 1378 (aromatic and heteroaromatic ring), 1277, 1210, 1198, 1153 (C-O-C, ester); ^1^H-NMR (500 MHz, CDCl_3_): δ 9.83 (1H, d, *J =* 8.5 Hz, H-10); 8.74 (1H, s, H-6); 7.90 (2H, q, *J =* 7.5, 8.5 Hz, H-7, H-9), 7.75 (1H, t, *J =* 7.5 Hz, H-8), 7.69 (1H, s, H-2), 7.61 (1H, dd, *J =* 1.5, 8.5 Hz, H-17), 7.55 (1H, d, *J =* 1.5 Hz, H-13), 6.92 (1H, d, *J =* 8.5 Hz, H-16), 4.84 (2H: CH_2_ of methyl acetate group from 15 position, s), 4.80 (2H: CH_2_ of methyl acetate group from 14 position, s), 3.94 (3H, s, CH_3_ of methoxycarbonyl group from 1 position), 3.83 (3H: CH_3_ of methyl acetate group from 15 position, s), 3.80 (3H: CH_3_ of methyl acetate group from 14 position, s), ^13^C-NMR (125 MHz, CDCl_3_): δ 183.1 (CO keto), 169.1 (CO keto ester from 15 position), 169.0 (CO keto ester from 14 position), 164.8 (CO keto ester from 1 position), 151.8 (C-15), 147.7 (C-14), 146.6 (C-6), 133.1 (C-9), 130.3 (C-10b), 129.9 (C-8), 127.7 (C-7), 127.6 (C-10), 127.1 (C-12), 127.0 (C-3), 126.9 (C-10a), 125.7 (C-17); 124.1 (C-2), 122.3 (C-6a), 115.9 (C-13), 113.5 (C-16), 107.8 (C-1), 66.4 (CH_2_ of methyl acetate group from 15 position, -CH_2_-COOMe), 66.3 (CH_2_ of methyl acetate group from 14 position, -CH_2_-COOMe), 52.5 (CH_3_ of methyl acetate group from 15 position, -CH_2_-COOMe), 52.4 (CH_3_ of methyl acetate group from 14 position, -CH_2_-COOMe), 52.0 (CH_3_ of methoxycarbonyl group from 1 position). Anal. calc. for C_26_H_22_N_2_O_9_ (506.47): C 61.66, H 4.38, N 5.53; found: C 61.58, H 4.34, N 5.46.

Dimethyl 2,2′-((5-(1-(methoxycarbonyl)pyrrolo[2,1-a]phthalazine-3-carbonyl)-1,3-phenylene)bis(oxy)) diacetate (**5e**). 0.375 g, 74% (under classical heating), 0.456 g, 90% (under microwave) and 0.451 g, 89% (under ultrasounds) as white crystals, m.p. 183–184 °C; R_f_ (98/2 CH_2_Cl_2_/CH_3_OH) 0.40; IR (cm^−1^): 3105, 3088, 3027 (C-H arom.), 2957 (C-H aliph.), 1742, 1706 (C=O, ester), 1647 (C=O, keto), 1539, 1519, 1470, 1389 (aromatic and heteroaromatic ring), 1287, 1251, 1989, 1115 (C-O-C, ester); ^1^H-NMR (500 MHz, CDCl_3_): δ 9.84 (1H, d, *J* = 8.5 Hz, H-10); 8.78 (1H, s, H-6); 7.91 (2H, m, overlapped peaks, H-7, H-9), 7.76 (1H, t, *J =* 7.5 Hz, H-8), 7.72 (1H, s, H-2), 7.10 (2H, d, *J =* 2.0 Hz, H-13, H-17), 6.76 (1H, t, *J =* 2.0, Hz, H-15), 4.68 (4H: CH_2_ of methyl acetate groups from 14 and 16 positions, s), 3.94 (3H, s, CH_3_ of methoxycarbonyl group from 1 position), 3.81 (6H: CH_3_ of methyl acetate groups from 14 and 16 positions, s), ^13^C-NMR (125 MHz, CDCl_3_): δ 183.5 (CO keto), 169.0 (CO keto ester from 14 and 16 positions), 164.7 (CO keto ester from 1 position), 159.0 (C-14, C-16), 146.7 (C-6), 141.4 (C-3), 133.2 (C-9), 130.8 (C-10b), 130.0 (C-8), 127.8 (C-7), 127.7 (C-10), 126.9 (C-12), 126.8 (C-10a), 125.2 (C-2); 122.4 (C-6a), 109.2 (C-13, C-17), 108.1 (C-1), 106.5 (C-15), 65.6 (CH_2_ of methyl acetate groups from 14 and 16 positions, -CH_2_-COOMe), 52.5 (CH_3_ of methyl acetate groups from 14 and 16 positions, -CH_2_-COOMe), 52.0 (CH_3_ of methoxycarbonyl group from 1 position). Anal. calc. for C_26_H_22_N_2_O_9_ (506.47): C 61.66, H 4.38, N 5.53; found: C 61.60, H 4.34, N 5.46.

Dimethyl 3-(2,4-bis(2-methoxy-2-oxoethoxy)benzoyl)pyrrolo[2,1-a]phthalazine-1,2-dicarboxylate (**6a**). 0.310 g, 55% (under classical heating), 0.463 g, 82% (under microwave) and 0.457 g, 81% (under ultrasounds) as yellow oil; R_f_ (98/2 CH_2_Cl_2_/CH_3_OH) 0.36; IR (cm^−1^): 3111, 3044, 3008 (C-H arom.), 2956 (C-H aliph.), 1738, 1706 (C=O, ester), 1629 (C=O, keto), 1546, 1522, 1501, 1455, 1387 (aromatic and heteroaromatic ring), 1279, 1239, 1209, 1162 (C-O-C, ester); ^1^H-NMR (500 MHz, CDCl_3_): δ 9.07 (1H, d, *J =* 8.0 Hz, H-10); 8.42 (1H, s, H-6); 7.76 (3H, m, H-17, H-7, H-9), 7.61 (1H, t, *J =* 7.5 Hz, H-8), 6.54 (1H, d, *J* = 9.0 Hz, H-16), 6.34 (1H, s, H-14), 4.63 (2H: CH_2_ of methyl acetate group from 15 position, s), 4.25 (2H: CH_2_ of methyl acetate group from 13 position, s), 3.91 (3H, s, CH_3_ of methoxycarbonyl group from 1 position), 3.80 (3H: CH_3_ of methyl acetate group from 15 position, s), 3.74 (3H: CH_3_ of methyl acetate group from 13 position, s), 3.69 (3H, s, CH_3_ of methoxycarbonyl group from 2 position), ^13^C-NMR (125 MHz, CDCl_3_): δ 183.2 (CO keto), 168.1 (CO keto ester from 15 position), 167.4 (CO keto ester from 13 position), 165.2 (CO keto ester from 2 position), 165.1 (CO keto ester from 1 position), 162.6 (C-15), 159.2 (C-13), 146.4 (C-6), 133.2 (C-9), 133.1 (C-17), 130.4 (C-10b), 129.1 (C-8), 128.6 (C-3), 128.0 (C-7), 126.8 (C-12), 125.5 (C-10), 125.2 (C-10a); 123.3 (C-2), 121.5 (C-6a), 106.8 (C-1), 106.5 (C-16), 101.0 (C-14), 66.4 (CH_2_ of methyl acetate group from 15 position, -CH_2_-COOMe), 65.4 (CH_2_ of methyl acetate group from 13 position, -CH_2_-COOMe), 53.7 (CH_3_ of methoxycarbonyl group from 1 position), 52.9 (CH_3_ of methyl acetate group from 15 position, -CH_2_-COOMe), 52.4 (CH_3_ of methyl acetate group from 13 position, -CH_2_-COOMe), 52.3 (CH_3_ of methoxycarbonyl group from 2 position). Anal. calc. for C_28_H_24_N_2_O_11_ (564.50): C 59.58, H 4.29, N 4.96; found: C 59.53, H 4.25, N 4.90.

Dimethyl 3-(2,5-bis(2-methoxy-2-oxoethoxy)benzoyl)pyrrolo[2,1-a]phthalazine-1,2-dicarboxylate (**6b**) 0.282 g, 50% (under classical heating), 0.435 g, 77% (under microwave) and 0.435 g, 77% (under ultrasounds) as yellow oil; R_f_ (98/2 CH_2_Cl_2_/CH_3_OH) 0.40; IR (cm^−1^): 3113, 3015, 2998 (C-H arom.), 2945, 2928 (C-H aliph.), 1739, 1708 (C=O, ester), 1618 (C=O, keto), 1540, 1497, 1448, 1429, 1387 (aromatic and heteroaromatic ring), 1299, 1285, 1222, 1176 (C-O-C, ester); ^1^H-NMR (500 MHz, CDCl_3_): δ 9.14 (1H, d, *J* = 8.0 Hz, H-10); 8.42 (1H, s, H-6); 7.79 (2H, q, *J* = 7.5, 8.0 Hz, H-7, H-9), 7.63 (1H, t, *J* = 7.5 Hz, H-8), 7.23 (1H, d, *J* = 3.0 Hz, H-17), 7.06 (1H, dd, *J* = 3.0, 8.0 Hz, H-15), 6.75 (1H, d, *J* = 8.0 Hz, H-14), 4.62 (2H: CH_2_ of methyl acetate group from 16 position, s), 4.25 (2H: CH_2_ of methyl acetate group from 13 position, s), 3.91 (3H, s, CH_3_ of methoxycarbonyl group from 1 position), 3.76 (3H: CH_3_ of methyl acetate group from 16 position, s), 3.71 (3H: CH_3_ of methyl acetate group from 13 position, s), 3.36 (3H, s, CH_3_ of methoxycarbonyl group from 2 position), ^13^C-NMR (125 MHz, CDCl_3_): δ 183.7 (CO keto), 169.1 (CO keto ester from 16 position), 168.3 (CO keto ester from 13 position), 165.1 (CO keto ester from 2 position), 164.9 (CO keto ester from 1 position), 152.7 (C-16), 152.1 (C-13), 146.4 (C-6), 133.1 (C-9), 130.1 (C-10b), 129.4 (C-8), 129.3 (C-3), 128.0 (C-7), 126.6 (C-12), 125.9 (C-10a), 125.8 (C-10); 122.8 (C-2), 121.6 (C-6a), 121.0 (C-15), 115.6 (C-17), 115.3 (C-14), 106.8 (C-1), 67.2 (CH_2_ of methyl acetate group from 16 position, -CH_2_-COOMe), 65.8 (CH_2_ of methyl acetate group from 13 position, -CH_2_-COOMe), 52.5 (CH_3_ of methoxycarbonyl group from 1 position), 52.4 (CH_3_ of methyl acetate group from 16 position, -CH_2_-COOMe), 52.3 (CH_3_ of methyl acetate group from 13 position, -CH_2_-COOMe), 51.9 (CH_3_ of methoxycarbonyl group from 2 position). Anal. calc. for C_28_H_24_N_2_O_11_ (564.50): C 59.58, H 4.29, N 4.96; found: C 59.52, H 4.24, N 4.89.

Dimethyl 3-(2,6-bis(2-methoxy-2-oxoethoxy)benzoyl)pyrrolo[2,1-a]phthalazine-1,2-dicarboxylate (**6c**). 0.271 g, 48% (under classical heating), 0.423 g, 75% (under microwave) and 0.423 g, 75% (under ultrasounds) as yellow oil; R_f_ (98/2 CH_2_Cl_2_/CH_3_OH) 0.36; IR (cm^−1^): 3099, 3051, 3011 (C-H arom.), 2955 (C-H aliph.), 1735, 1708 (C=O, ester), 1630 (C=O, keto), 1543, 1499, 1457, 1398, 1385 (aromatic and heteroaromatic ring), 1277, 1239, 1210, 1162 (C-O-C, ester); ^1^H-NMR (500 MHz, CDCl_3_): δ 9.44 (1H, d, *J* = 8.5 Hz, H-10); 8.53 (1H, s, H-6); 7.82 (2H, m, H-7, H-9), 7.69 (1H, t, *J* = 8.0 Hz, H-8), 7.28 (1H, t, *J* = 8.5 Hz, H-15), 6.50 (2H, d, *J* = 8.5 Hz, H-14, H-16), 4.52 (4H: CH_2_ of methyl acetate groups from 13 and 17 positions, s), 3.88 (3H, s, CH_3_ of methoxycarbonyl group from 1 position), 3.73 (3H, s, CH_3_ of methoxycarbonyl group from 2 position), 3.56 (6H: CH_3_ of methyl acetate groups from 13 and 17 positions, s), ^13^C-NMR (125 MHz, CDCl_3_): δ 181.3 (CO keto), 169.0 (CO keto ester from 13 and 17 positions), 166.1 (CO keto ester from 2 position), 164.2 (CO keto ester from 1 position), 157.1 (C-13, C-17), 146.5 (C-6), 133.1 (C-9), 131.6 (C-15), 131.5 (C-10b), 129.9 (C-8), 127.7 (C-7), 127.6 (C-3), 127.1 (C-2), 126.8 (C-10), 126.5 (C-10a), 122.1 (C-6a), 120.8 (C-12), 106.8 (C-14, C-16), 106.6 (C-1), 66.7 (CH_2_ of methyl acetate groups from 13 and 17 positions, -CH_2_-COOMe), 52.7 (CH_3_ of methoxycarbonyl group from 1 position), 52.3 (CH_3_ of methoxycarbonyl group from 2 position), 52.1 (CH_3_ of methyl acetate groups from 13 and 17 positions, -CH_2_-COOMe). Anal. calc. for C_28_H_24_N_2_O_11_ (564.50): C 59.58, H 4.29, N 4.96; found: C 59.50, H 4.25, N 4.91.

Dimethyl 3-(3,4-bis(2-methoxy-2-oxoethoxy)benzoyl)pyrrolo[2,1-a]phthalazine-1,2-dicarboxylate (**6d**). 0.294 g, 52% (under classical heating), 0.463 g, 82% (under microwave) and 0.452 g, 80% (under ultrasounds) as yellow oil; R_f_ (98/2 CH_2_Cl_2_/CH_3_OH) 0.40; IR (cm^−1^): 3101, 3049, 3005 (C-H arom.), 2952 (C-H aliph.), 1739, 1711 (C=O, ester), 1623 (C=O, keto), 1546, 1519, 1499, 1439, 1371 (aromatic and heteroaromatic ring), 1277, 1245, 1201, 1145 (C-O-C, ester); ^1^H-NMR (500 MHz, CDCl_3_): δ 8.70 (1H, d, *J* = 8.0 Hz, H-10); 8.45 (1H, s, H-6); 7.82 (2H, q, *J* = 7.5, 8.0 Hz, H-7, H-9), 7.63 (1H, t, *J* = 7.5 Hz, H-8), 7.50 (1H, s, H-13), 7.33 (1H, d, *J* = 8.5 Hz, H-17), 6.77 (1H, d, *J* = 8.5 Hz, H-16), 4.78 (2H: CH_2_ of methyl acetate group from 15 position, s), 4.76 (2H: CH_2_ of methyl acetate group from 14 position, s), 3.98 (3H, s, CH_3_ of methoxycarbonyl group from 1 position), 3.77 (3H: CH_3_ of methyl acetate group from 15 position, s), 3.76 (6H: CH_3_ of methyl acetate group from 14 position, s), 3.58 (3H, s, CH_3_ of methoxycarbonyl group from 2 position), ^13^C-NMR (125 MHz, CDCl_3_): δ 185.5 (CO keto), 168.8 (CO keto ester from 2 position), 168.7 (CO keto ester from 15 position), 168.6 (CO keto ester from 14 position), 168.1 (CO keto ester from 1 position), 166.0 (C-15), 163.8 (C-14), 147.2 (C-6), 133.5 (C-9), 132.0 (C-10b), 129.1 (C-8), 128.5 (C-12), 128.4 (C-7), 126.7 (C-3), 125.9 (C-2), 124.5 (C-10); 124.2 (C-10a), 121.2 (C-17), 118.8 (C-6a), 114.3 (C-13), 113.2 (C-16), 107.6 (C-1), 66.2 (CH_2_ of methyl acetate group from 15 position, -CH_2_-COOMe), 66.0 (CH_2_ of methyl acetate group from 14 position, -CH_2_-COOMe), 52.8 (CH_3_ of methoxycarbonyl group from 1 position), 52.5 (CH_3_ of methyl acetate group from 15 position, -CH_2_-COOMe), 52.3 (CH_3_ of methyl acetate group from 14 position, -CH_2_-COOMe), 52.2 (CH_3_ of methoxycarbonyl group from 2 position). Anal. calc. for C_28_H_24_N_2_O_11_ (564.50): C 59.58, H 4.29, N 4.96; found: C 59.52, H 4.24, N 4.89.

Dimethyl 3-(3,5-bis(2-methoxy-2-oxoethoxy)benzoyl)pyrrolo[2,1-a]phthalazine-1,2-dicarboxylate (**6e**). 0.294 g, 52% (under classical heating), 0.452 g, 80% (under microwave) and 0.440 g, 78% (under ultrasounds) as yellowish crystals, m.p. 138–139 °C; R_f_ (98/2 CH_2_Cl_2_/CH_3_OH) 0.40; IR (cm^−1^): 3098, 3044, 3001 (C-H arom.), 2953 (C-H aliph.), 1740, 1706 (C=O, ester), 1635 (C=O, keto), 1550, 1531, 1497, 1477, 1389 (aromatic and heteroaromatic ring), 1285, 1262, 1177, 1114 (C-O-C, ester); ^1^H-NMR (500 MHz, CDCl_3_): δ 8.64 (1H, d, *J* = 8.5 Hz, H-10); 8.41 (1H, s, H-6); 7.72 (2H, m, overlapped peaks, H-7, H-9), 7.57 (1H, t, *J* = 7.5 Hz, H-8), 6.95 (2H, d, *J* = 2.0 Hz, H-13, H-17), 6.70 (1H, t, *J* = 2.0 Hz, H-15), 4.57 (4H: CH_2_ of methyl acetate groups from 14 and 16 positions, s), 3.93 (3H, s, CH_3_ of methoxycarbonyl group from 1 position), 3.71 (6H: CH_3_ of methyl acetate groups from 14 and 16 positions, s), 3.55 (3H, s, CH_3_ of methoxycarbonyl group from 2 position), ^13^C-NMR (125 MHz, CDCl_3_): δ 185.9 (CO keto), 168.6 (CO keto ester from 14 and 16 positions), 165.5 (CO keto ester from 2 position), 163.7 (CO keto ester from 1 position), 158.9 (C-14, C-16), 147.1 (C-6), 139.7 (C-3), 133.3 (C-9), 129.1 (C-8), 128.3 (C-7), 127.8 (C-12), 126.3 (C-10b), 124.5 (C-2), 124.4 (C-10), 121.1 (C-10a); 119.6 (C-6a), 108.4 (C-13, C-17), 107.6 (C-15), 107.4 (C-1), 65.2 (CH_2_ of methyl acetate groups from 14 and 16 positions, -CH_2_-COOMe), 52.6 (CH_3_ of methoxycarbonyl group from 1 position), 52.2 (CH_3_ of methyl acetate groups from 14 and 16 positions, -CH_2_-COOMe), 52.1 (CH_3_ of methoxycarbonyl group from 2 position). Anal. calc. for C_28_H_24_N_2_O_11_ (564.50): C 59.58, H 4.29, N 4.96; found: C 59.51, H 4.24, N 4.90.

## 4. Conclusions

A thorough study concerning the Huisgen [3 + 2] dipolar cycloadditions of phthalazinium ylides to methyl propiolate or dimethyl acetylenedicarboxylate was presented here. The reaction pathway is straightforward and efficient and allows synthesis of pyrrolophthalazine derivatives, compounds which are difficult to be obtained otherwise. A comparative study concerning the efficiency of synthesis, conventional thermal heating (TH) versus microwave (MW) and ultrasound (US) irradiation, has been performed. The cycloaddition reactions of phthalazinium ylides to methyl propiolate occur regiospecific, with a single regioisomer being obtained, the one in which the bond is formed between the ylide carbon and the non-substituted carbon atom of the methyl propiolate, which means that the reaction is under a charge control. Under conventional TH, the cycloaddition reaction of phthalazinium ylides with DMAD occurred to a mixture of inseparable partial and fully aromatized pyrrolophthalazine derivatives. Under MW or US irradiation, the Huisgen [3 + 2] dipolar cycloaddition reactions of ylides to DMAD became selective, with only fully aromatized compounds being obtained. As for the mechanism, the formation of fully aromatized compounds could be explained by an oxidative dehydrogenation of the partial aromatized intermediaries, which are leading to the more thermodynamically stable compounds. Besides selectivity, it has to be noticed that the reaction setup under MW or US irradiation offers a number of certain other advantages: higher yields, decreasing of the amount of used solvent comparative with TH, decreasing of the reaction time from hours to minutes and decreasing of the consumed energy. Taking into consideration the above-mentioned advantages, the Huisgen [3 + 2] dipolar cycloaddition reactions of ylides to various dipolarophiles under MW or US irradiation could be considered environmentally friendly.

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
