# Peer review of "Huisgen [3 + 2] Dipolar Cycloadditions of Phthalazinium Ylides to Activated Symmetric and Non-Symmetric Alkynes"

_molecules, 2020, doi:10.3390/molecules25194416_

Round 1

Reviewer 1 Report

Mangalagiu, Zbancioc, and their coworkers present a straightforward and efficient pathway for the synthesis of pyrrolophthalazine derivatives via the Huisgen [3+2] dipolar cycloaddition of phthalazinium ylides and methyl propiolate or
dimethyl acetylenedicarboxylate (DMAD). A thoroughly comparative study on the efficiency of synthesis by conventional thermal heating (TH) versus microwave (MW) and ultrasound (US) irradiations have been performed. The cycloaddition
occurs regiospecifically, affording single regioisomers. Under conventional TH, the cycloaddition of phthalazinium ylides with DMAD gives a mixture of inseparable partial and fully aromatized pyrrolophthalazine cycloadducts, while MW or US irradiation leads only to fully aromatized compounds, the reactions becoming selective. A feasible mechanism for formation of fully aromatized compounds is assumed. Besides selectivity, the reaction setup under MW or US irradiation offer some advantages: higher yields, decreasing of the solvent amount, reaction time, and energy consumption comparative with TH, resulting in the reactions environmentally friendly.

Comments

  1. The authors mentioned that MW and US irradiations can accelerate the reaction rates. However, they conducted the reaction at different reaction temperatures under TH, MW, and US. They should compare the outcomes of the reactions at the same temperatures under different heating conditions. On the influence of MW irradiation on the selectivities in organic reactions, please see the reviews and they should be cited. 1) Microwave irradiation and selectivities in organic reactions. Prog. Chem. 2007, 19(5), 700–712. 2) Selectivities in the Microwave-Assisted Organic Reactions. In Banik, Bimal K.; Bandyopadhyay, Debasish Eds. Advances in Microwave Chemistry. Chapter 6, 257-291,  The different reaction rates maybe attributed to the different reaction temperatures. For the example, please see: Effects of the microwave power on the microwave assisted esterification. Curr. Microw. Chem. 2017, 4(2), 158–162.
  2. On the aromitization from 7 to 6, the authors should conduct some control experiments, making sure their conclution. For example, conduct their reaction under N2, conduct a reaction with 7 as starting material in the presence of O2. On the other hand, -[2H+] is incorrect, possibly, should be + O2, -H2O. It should be verified by the control experiments.
  3. In scheme 3, the structure of 2a-e is incorrect, CH2  should be CH. The transformation from 7' to 7 should be TEA-catalyzed process, in which TEA first abstracts the proton, double bond shifts, and them TEAH+ transfers the proton to the generated carbanion.

Author Response

  1. The authors mentioned that MW and US irradiations can accelerate the reaction rates. However, they conducted the reaction at different reaction temperatures under TH, MW, and US. They should compare the outcomes of the reactions at the same temperatures under different heating conditions.

By the acceleration of the reaction rates we mean that the reaction time decrease, we did not intend to study the kinetic of these reactions. Our intention was to find the best results obtained with optimized reaction conditions under MW and US irradiation and to compare with the best results obtained under optimized reaction conditions used in thermal heating. We have added these remarks in the manuscript (See line 129-133).

On the influence of MW irradiation on the selectivities in organic reactions, please see the reviews and they should be cited. 1) Microwave irradiation and selectivities in organic reactions. ProgChem. 200719(5), 700–712. 2) Selectivities in the Microwave-Assisted Organic Reactions. In Banik, Bimal K.; Bandyopadhyay, Debasish Eds. Advances in Microwave Chemistry. Chapter 6, 257-291,  The different reaction rates maybe attributed to the different reaction temperatures. For the example, please see: Effects of the microwave power on the microwave assisted esterification. Curr. Microw. Chem20174(2), 158–162.

We have added these references (See ref 29-31) in the manuscript and discussed the influence of the MW and US over the acceleration of the reaction rates (See line 140-159).

  1. On the aromitization from 7 to 6, the authors should conduct some control experiments, making sure their conclution. For example, conduct their reaction under N2, conduct a reaction with 7 as starting material in the presence of O2. On the other hand, -[2H+] is incorrect, possibly, should be + O2, -H2O. It should be verified by the control experiments.

The oxidative dehydrogenation of the 7a-e to 6a-e cycloadducts occurs in the reaction mixture both before and after the workup of the reaction mixture during separation (we have added this information in the manuscript – see line 108-110).

The aromatization of various dihydropyrrolo to pyrrolo cycloadducts make the subject of a different paper that we have prepared for publication.

  1. In scheme 3, the structure of 2a-e is incorrect, CH2  should be CH.

We have made the correction, thank you for observation (See scheme 3).

The transformation from 7' to 7 should be TEA-catalyzed process, in which TEA first abstracts the proton, double bond shifts, and them TEAH+ transfers the proton to the generated carbanion.

We have introduced the mechanism of the isomerization of the 7’ to 7 according your observation (see line 102-108).

Reviewer 2 Report

The work entitled "Huisgen 3 + 2 Dipolar Cycloadditions of Phthalazinium Ylides to Activated Symmetric and Non-Symmetric Alkynes" by Vasilichia Antoci, Costel Moldoveanu, Ramona Danac, Violeta Mangalagiu, and Gheorghita Zbancioc present an interesting work on enhancing of yield and selectivity by ultrasounds and microwaves in Huisgen 3 + 2 dipolar cycloadditions on a selected model. Generally, the work is well designed, although represents a narrow group of chemicals. While the discussion is modest, the work may anyway have a practical value.

Here are my comments.

Authors tend to use references in a very general form. Numerous works, for example [4-12], [13-21], [27-31] are cited in the introduction giving only a vague feeling of the topics. Therefore, I would suggest reducing theses general citations or make it more specific, but also to deepen the discussion and refer more to literature in the Result and discussion section.

I would also suggest introducing a brief explanation of how ultrasounds and microwaves stimulate to process to decrease the reaction time from a week to minutes.

I have not found any explanations of how the stereoisomers 5a-5e were assigned to the structure.

Since NMR and IR spectra description constitutes 25% of the work I would suggest moving it to the supplementary materials leaving only the general procedures in the manuscript. Alternatively, an advisable solution it to deposit the data in an external site as described at https://www.mdpi.com/journal/molecules/instructions#suppmaterials

It is also not clear what was the oxygen impact of the dehydration in thermal condition leading to aromatization (7 -> 6).

I do not recommend this for publication unless these points will be addressed. Therefore, my recommendation is a major revision.

Author Response

Authors tend to use references in a very general form. Numerous works, for example [4-12], [13-21], [27-31] are cited in the introduction giving only a vague feeling of the topics. Therefore, I would suggest reducing theses general citations or make it more specific, but also to deepen the discussion and refer more to literature in the Result and discussion section.

We have made the correction, thank you for observation (See line 35-36, 41-42 and 54).

I would also suggest introducing a brief explanation of how ultrasounds and microwaves stimulate to process to decrease the reaction time from a week to minutes.

We have added new references (See ref 29-31) in the manuscript and discussed the influence of the MW and US over the acceleration of the reaction rates (See line 140-159).

I have not found any explanations of how the stereoisomers 5a-5e were assigned to the structure.

We explained the assignment of the structure for the stereisomers 5a-e (based on the spectral analysis-NMR). (See line 72-74 and 89-94).

Since NMR and IR spectra description constitutes 25% of the work I would suggest moving it to the supplementary materials leaving only the general procedures in the manuscript. Alternatively, an advisable solution it to deposit the data in an external site as described at https://www.mdpi.com/journal/molecules/instructions#suppmaterials

Since the General considerations from Manuscript preparation of the Molecules Journal regarding the Experimental data state that “Complete characterization data must be given for all new compounds.”, and since in our previous papers we have included this part in the manuscript, we have prepared this manuscript in the same manner. However, we leave it to the editor to decide whether or not to keep this part in the manuscript.

It is also not clear what was the oxygen impact of the dehydration in thermal condition leading to aromatization (7 -> 6).

The oxidative dehydrogenation of the 7a-e to 6a-e cycloadducts occurs in the reaction mixture both before and after the workup of the reaction mixture during separation (we have added this information in the manuscript – see line 108-110).

The aromatization of various dihydropyrrolo to pyrrolo cycloadducts make the subject of a different paper that we have prepared for publication.

Round 2

Reviewer 1 Report

The authors addressed almost all comments. However, the following two comments need to be further improved.

1.  On the aromitization from 7 to 6, the authors should conduct some control experiments, making sure their conclution. For example, conduct their reaction under N2, conduct a reaction with 7 as starting material in the presence of O2. On the other hand, -[2H+] is incorrect, possibly, should be + O2, -H2O. It should be verified by the control experiments. The control experiments should be added

2. Scheme 3 is not updated, please update it as the authors' responses. That is, from 7 to 6, it should be O2/-H2O, rather than -2H.

3.  Reference 30 should be Xu, J. Selelectivities in microwave-assisted organic reactions. In Banik, B.K, Bandyopadhyay, D. Eds. Advances in.........

Author Response

  1. On the aromitization from 7 to 6, the authors should conduct some control experiments, making sure their conclution. For example, conduct their reaction under N2, conduct a reaction with 7 as starting material in the presence of O2. On the other hand, -[2H+] is incorrect, possibly, should be + O2, -H2O. It should be verified by the control experiments. The control experiments should be added

Indeed, the aromatization process of the compounds 7a-e in 6a-e takes place in the presence of a hydrogen acceptor which is most often oxygen from the air. There are situations in which the aromatization can be accelerated by the use of oxidizing agents (eg TPCD as in the following references [Dumitrascu, F; Draghici, C; Caproiu, M.T.; Dumitrescu, D.; Badoiu, A. New pyrrolo[2,1-a]phthalazine derivatives by 1,2-dipolar cycloaddition reactions Rev. Roum. Chim. 2006, 51(7-8), 643–647. Zhou, J.; Hu, Y.; Hu, H. Synthesis of pyrrolo[2,1‐a]phthalazines by 1,3‐dipolar cycloaddition of phthalazinium N‐ylides with alkenes in the presence of tetrakis‐pyridine cobalt (II) dichromate J. Het. Chem. 2000, 37, 1165-1168]). The N ylides and their reactions is an area of tradition in our research group, professors Zugravescu and Petrovanu laid the groundwork of this domain since the 60s. Such experiments to control the aromatization process have been done before (Mangalagiu, G.; Mangalagiu, I.; Olariu, R; Petrovanu, M. 4-Methylpyrimidinium ylides II: Selective reactions of pyrimidinium ylides with activated alkyne Synthesis 2000, 14, 2047-2050. Mangalagiu. I; Mangalagiu, G.; Deleanu, C.; Drochioiu, G.; Petrovanu, M. 4-Methylpyrimidinium ylides. Part 7: 3+2 Dipolar cycloadditions to non-symmetrical substituted alkenes and alkynes Tetrahedron 2003, 59(1), 111-114) the reactions being performed in the nitrogen, air, oxygen or even carbon dioxide atmosphere. These experiments really showed that aromatization is delayed or even stopped in the nitrogen atmosphere and accelerated in the oxygen atmosphere. As a result, we modified in Scheme 3 the reaction conditions of the last stage according to your indications, and explained in the manuscript the role of the oxygen in the aromatization process (See lines 107-112 and related references).

In the present case, such experiments are not justified except from a theoretical point of view: establishing the mechanism, a mechanism that is already known in the literature. However, partially hydrogenated compounds cannot be isolated because the workup of the reaction mixture and the compounds purification (chromatography) is done in an air atmosphere. Moreover, the compounds are partially hydrogenated thus they do not have an aromatic pyrrole ring, they do not have a planar structure and therefore they do not have eventual optical properties (fluorescence).

  1. Scheme 3 is not updated, please update it as the authors' responses. That is, from 7 to 6, it should be O2/-H2O, rather than -2H.

We modified in Scheme 3 the reaction conditions of the last stage according to your indications (See Scheme 3).

  1. Reference 30 should be Xu, J. Selelectivities in microwave-assisted organic reactions. In Banik, B.K, Bandyopadhyay, D. Eds. Advances in.........

We have made the correction, thank you for observation (See line 500-501).

Reviewer 2 Report

The Authors have addressed all the issues I rose in my initial review. I have also noticed that the changes enhanced the scientific level of the work due to a deeper discussion and increasing the amount of evidence and thus the credibility of the reported results.

Therefore, I recommend publication of the work in its present form.

Author Response

The Authors have addressed all the issues I rose in my initial review. I have also noticed that the changes enhanced the scientific level of the work due to a deeper discussion and increasing the amount of evidence and thus the credibility of the reported results.

Therefore, I recommend publication of the work in its present form.

Thank you for your appreciation.